# Evaluation of Methodologies and Measures on the Usability of Social Robots: A Systematic Review

Minjoo Jung [1], May Jorella S. Lazaro [2] and Myung Hwan Yun [1],*

1 Department of Industrial Engineering, Seoul National University, Seoul 08826, Korea; mjjung03@snu.ac.kr
2 Interdisciplinary Program in Cognitive Science, Seoul National University, Seoul 08826, Korea; ellalazaro@snu.ac.kr
* Correspondence: mhy@snu.ac.kr

**Abstract:** This paper systemically reviews and clarifies the state-of-the-art HRI evaluation studies especially for the usability of social robots. A total of 36 articles were collected through a keyword, abstract, title search from various online search engines. Afterwards, 163 measures were selected and reviewed carefully. This research was classified into two parts. In the first part, evaluation methodologies were investigated according to (1) type of stimuli on evaluation, (2) evaluation technique, and (3) criteria of participants. In the second part, assessment measures were collected and the model of attitude towards a social robot is proposed. As a result, this study suggests practical strategies for selecting appropriate methods and measures that meet specific requirements of research. The proposed hierarchical structure of assessment measures is expected to contribute to both practical use and academic use.

**Keywords:** social robots; usability evaluation methods; usability dimensions; human-robot interaction





## 1. Introduction

Around 1495, Leonardo da Vinci had a vision and designed a robot that has arms and movable joints. After 500 years, we have almost 30 million active robots in the world [1]. In 1979, the Robot Institute of America defined a robot as a reprogrammable and a multifunctional manipulator designed to move material, parts, tools or specialized devices. The International Federation of Robotics classifies robots into two categories, industrial robot and service robot. A service robot can be either a professional service robot or a personal service robot. A personal service robot often called a social robot, is on a big uptake. The professional robot is usually used for commercial task and its task is clearly defined. However, a social robot is defined as a new type of robot and its main goal is social interaction with human [2,3] defined a social robot as a robot designed to evoke meaningful social interaction with its users. Thus, it can be concluded that social robot is different from professional robots in such a way that it does not necessarily need to accomplish a single fixed task, but rather, social robots are expected to express its emotions, to deliver complex dialogue, to learn, to understand natural cues, and to develop personality and social competencies [4–6].

In terms of numbers, a social robot accounts for a relatively small percentage of sales. It is only 4% of the total robot sales from 2015 to 2018 [1]. However, even though the expected number is relatively small, a social robot is the next big thing to the public. Since robotic technology is growing rapidly, people expect a social robot to fill a lack of human resources. A typical example of the role that people expect from a social robot is supporting children with special needs or aged population. Moreover, the idea that assuming the role of social robots as an assistive technology to change people's attitudes and behavior is stimulating researchers' interest [7,8].

On the contrary to these interests, there are still many barriers to adopt a social robot to consumers' life such as its performance and unfamiliarity. To overcome those barriers,

there are numerous studies to evaluate consumers' attitude toward a social robot for improvement. However, practitioners in the industry still find it difficult to understand evaluation methods and to select appropriate measures from scattered previous studies.

## 2. Research Method

### 2.1. Research Objectives

Robot industry has been steadily growing since after the first industrial robots developed in 1960s. At that time, the most appealing role of robots is replacing human beings to do unreachable by or unsafe job. For those kinds of robots, there are clear evaluation indices like task completion rate and time. However, we can't apply those indices for evaluation of social robots. When it comes to social robot, their task is ambiguous and everyone has different expectations for the role of that. We still can apply traditional evaluation indices in HCI such as ease of use, usefulness. However, usability is basic concept and further research is needed on what makes social robots attractive. In this paper, we will suggest main pillars affecting attitude towards social robots, and readers can easily understand what's important in designing a social robot to make it more attractive.

One more thing that is difference from the traditional HCI research is evaluation methodology. In the traditional HCI research area, most of elements that we want to evaluate are visual components and there are various prototyping tools to make visual stimuli easily for quick and dirty research. However, it is still difficult to develop working prototype which can be executed like final commercial product and practitioners constantly needed ideas how to evaluate social robots in early stage of developing process. In order to help them, our analysis is guided by these following research questions.

RQ1. What kind of usability evaluation methodologies are deployed in social robot studies?

- To provide a structural taxonomy of evaluation methods available
- To serve a guideline for identifying appropriate UX evaluation method for practitioners at the moment

RQ2. What are the main evaluation dimensions in social robot studies?

- To classify usability evaluation measures addressed in recent studies related to social robot evaluation

To concentrate on these research questions, we classify and summarize the state-of-the-art research relevant for social robot usability. The remainder of this article is structured as follows: First, we illustrate the methodology to select and extract the articles to be reviewed. In the second part, we address each research question based on the literature review. Finally, we conclude the article with brief summaries and remarks.

### 2.2. Search Methods and Selection Criteria

In this review, articles were extracted with a systemic approach. Four major online search engines were used to search for articles that can cover both engineering and psychological topics. Inclusion and screening criteria follow.

Search database. In this study, various online databases were searched including ScienceDirect, Web of Science, EBSCO and Scopus to provide a broad spectrum perspective.

Search keywords. At first, "social robot interaction" is used as a keyword for searching, and there was a substitutional term of "social robot", "socially interactive robot". To extend the coverage of this research, the keywords used in search engines were combinations of either "social robot" or "socially interactive robot" with "interaction" finally. After conducting the keyword search, 1985 papers were searched with the keywords "social robot" and "interaction". In sequence, 139 papers were found with the keywords "socially interactive robot" and "interaction".

Publication year. There was no limitation of publication year for selection because most of the articles related to social robot were published since 2000.

Publication type. This study covers only journal, conference articles and books. We didn't include other publication forms such as unpublished papers, masters and

doctoral dissertations, newspapers, patent. Journals, conferences articles and books were chosen because of its accessibility for both practitioners and academicians.

*2.3. Data Extraction and Synthesis*

After the selection and screening process, 36 studies remained for the review from the initial 613 searched papers. The selected articles were carefully reviewed and utilized to investigate experiment design and evaluation technique. The following topics were found through reading the full-text of the final chosen articles.

- Target audience of the robot: Elderly, Children, People who have a specific disease, Ordinary people
- Type of stimuli for evaluation: Text, Image, Video, Live interaction
- Evaluation technique: Questionnaire, Biometrics, Video analysis, Interview
- Main modality of robot interaction: Vision, Audition, Tactition, Thermoception

Based on these topics, data were extracted from 36 studies. However, data related to the main modality of robot interaction were not sufficient to draw a meaningful conclusion and the topic was excluded for this research.

## 3. Results

*3.1. Search Outcome*

3.1.1. Search Process

After removing 46 duplicated articles from initial candidates, 109 more articles related to the development of social robots in computer science or mechanics perspectives were excluded. Next, remained 458 articles are reviewed to select those to answer the purpose of this research and 109 articles were filtered by published journal and its titles. Finally, by reading the abstract of these articles, 36 articles are included in this study. Figure 1 shows the flow diagram of the selection and screening process used this study.

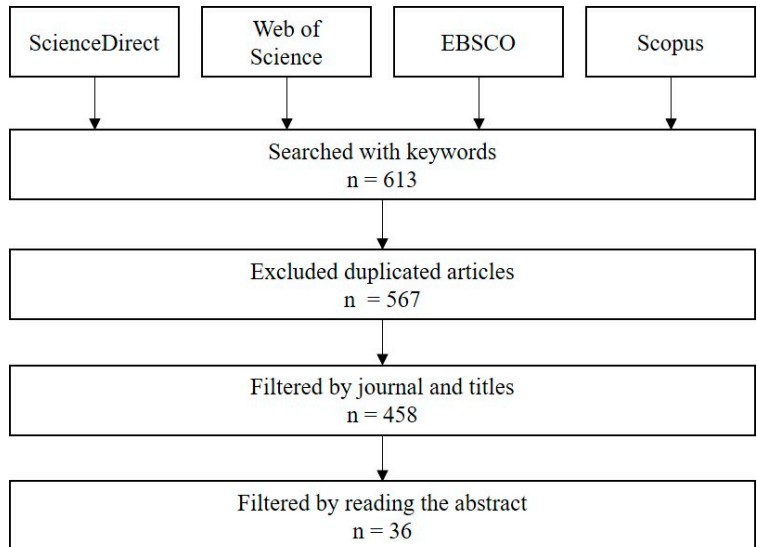

**Figure 1.** Flow diagram of selection and screening process.

3.1.2. Overview of Included Studies

From the 36 studies that we reviewed; characteristics of experiments design were investigated.

Design of experiments process. There were 3 studies which had two or more robots to compare for evaluation. In other studies, researchers wanted to prove their concept with an existing prototype in the early stage of robot development. In addition, 3 research that used static-group comparison were found. These studies aimed to verify the effectiveness

of the robot for particular target participants, and for the experiment, the target user (i.e., children with autism) and typical user (i.e., children without autism) were selected for group-comparison.

Participants. The average number of participants among 36 studies was 48.3 with a minimum of 3 and a maximum of 174. Some studies recruited a very small number of participants and their target user was people who have a specific disease.

### 3.2. Evaluation Methods

To address RQ1, this study classified usability evaluation methods in terms of (1) type of stimuli on evaluation, (2) evaluation technique and (3) criteria of participants.

### 3.2.1. Type of Stimuli

Generally, it is best to provide a product which has already completed the development process to the users and use it as a stimulus for evaluation. It is helpful to improve a product when the users are able to give feedback after using a final product in their natural environment. However, in the case of social robots, it is inefficient to evaluate the product after its actual physical development in consideration of its long and complicated development process. When we want to evaluate a product that is still in the stage of development, we provide users with various type of stimulus that can make up for unimplemented features of the product. Considering the particularity of this field, this paper analyzed which type of stimuli are usually provided and evaluated in the assessment, and what are the pros and cons of each type.

Among the 36 articles reviewed in this study, we found four types of stimuli used in evaluating the usability social robots: (1) a textual description of the robot, (2) a still image showing the robot, (3) a video showing the appearance and interaction of robot, (4) live interaction between the participant and the robot. Founded types of stimuli are summarized in Table 1.

**Table 1.** Summary of stimuli type.

| Type of Stimuli | Description | Count | Ratio |
|---|---|---|---|
| Text | A text stimulus can be a short description summarizing the product concept. | 1 | 3% |
| Image | An image stimulus is used to show the appearance of product in perspective. | 4 | 10% |
| Video | A video stimulus is more realistic and dynamic version of storyboard to communicate a possible use case clearly. | 4 | 10% |
| Live Interaction | Giving a working prototype or a finished product to user\to interact with freely. | 31 | 77% |

Text. There was only one study that used a stimulus delivered in the textual description and it was illustrated with photographs of the robot [9]. They used a text stimulus for comparing the effect of scenario media. It is difficult to describe the appearance of a robot in text, so a text scenario could be more appropriate to deliver the function of a robot but not its appearance. It is also mentioned in [9] that a text scenario is effective when evaluating the robot's functionality and users' general attitude toward a robot.

Images. There was four research that used a still image as a stimulus for evaluation [10–13]. Those research aimed to evaluate the robot's appearance to identify appropriate design or facial expression of robots. Images are relatively efficient in terms of time and cost, and it can also be designed for a specified purpose. However, in using images, it is hard to fully grasp different experiences of interaction between the user and a robot. In recent research, an image that represents a robot can't be understood by people as being with the subject in a shared context and it yields different behavioral responses than

physical robots in real-world [7]. Image is static and can't describe the delicate difference among robots' movements or changes of facial expression. Taking this limitation into account, using an image as a stimulus for social robot evaluation can only be effective if the subject for evaluation is the users' first impression towards the appearance of the robot.

Video. In many research, a video analysis was chosen commonly and was shown to be an effective research method to meet the goal of understanding user's perceptions of robotic technologies [10]. Despite known limitations [14], a video stimulus has advantages, including consistent experience across participants.

In addition, early studies which used videos proved the advantage of higher degrees of freedom in experimental control: for example, when overlaying the same behavior on the appearance of different gender [15] or neutralizing cultural effects by using standardized animation prototypes [16].

There were four studies which conducted the social robot evaluation through showing a video which contains the interaction between users and robots [9,10,12,16]. One of these studies [16] chose a video stimulus for conducting their survey via online. A video stimulus through an online survey could be an efficient substitution of interaction between subjects and a robot in a real-world setting. Similarly, in a study which provided VR modalities as stimuli to subjects which conveyed various posed robots, it was found that there is a significant effect between a physically present robot and a VR robot on both valence and arousal ratings.

However, it depends on which phase of development the robot is in. If the robot can't guarantee the best condition to interact with subject freely, we can use a video stimulus in place of a physically present robot to avoid usability challenges [10]. A video type can depict the context of use more accurately and the movement or facial expression of the robot more vividly than a static image.

Live interaction. 31 studies we reviewed provide a physically present robot to subjects for their research. Previous research found that physically present robots had been found to be more persuasive, arousing, and to be received more positively when compared to robots depicted in the video, images, or as virtual agents [12].

There were studies that used commercialized robots (e.g., Pepper, iCat, Karotz, Aibo, RoboVie) for their research [17–24]. When fully commercialized robots can be provided, test participants can interact with the robot freely. Especially [23] provided Karotz to participants, and they tried to use Karotz willingly in their home for a long time. This research was well designed to understand how subjects' beliefs on robots can be built. However, when researchers provided prototype robots as stimuli, 22.6% of studies chose a Wizard of Oz method for their interaction [9,20,22,25–28]. Even though a participant considers he or she communicates with the robot in real-time, a hidden experimenter controls the robot after a test participant commands something to the robot. This method can prevent unexpected errors of robots and make up for specific functionality which is not implemented yet.

### 3.2.2. Evaluation Technique

We found four types of evaluation techniques from reviewed articles: (1) questionnaires, (2) interview, (3) video analysis, (4) biometrics. Details of techniques were summarized in Table 2.

Questionnaires. There were 29 studies that used a questionnaire to evaluate HRI. Traditionally it is the most common instrument to evaluate UX so there are various verified questionnaires deployed in industry and academia. When we use a questionnaire to evaluate, it is easy to analyze and show its result in the statistics to others. However, when we want to understand users more behind its answer to a questionnaire, we cannot find any reason from the evaluation. That's why researchers do an interview or conduct any other type of research as a complementary way of a questionnaire.

**Table 2.** Summary of evaluation techniques.

| Evaluation Technique | Description | Count | Ratio |
|---|---|---|---|
| Questionnaires | Easy to gather a large amount of data and easy to analyze statistically.<br>Cannot interact with respondents and hard to interpret respondents' answer fully. | 29 | 62% |
| Video analysis | Good to observe and understand users behavior in natural setting<br>Needed much time and effort to analyze the video | 10 | 21% |
| Interview | Can ask more questions after users' answers to understand their intention clearer<br>Dependent on an interviewers' ability | 6 | 13% |
| Biometrics | Can get objective data from users directly<br>Hard to control unintended fluctuation of users data | 2 | 4% |

In this study, we found many researchers used the questionnaires on users' acceptance of the new product, a social robot which was verified in other previous studies already. They used one specific existing questionnaire or they combine two or more existing questionnaire to meet their research goal. We will introduce some of user acceptance model which is referred to in several studies in the next part of this article.

Interview. For conducting an interview, an interviewer can interact with the participant directly or can do telephone interviews. Notably, it is founded that face-to-face interviews were a complementary way of other evaluation techniques and its result was reported as additional opinion with their quantified rating. However, half of the research that conducted an interview targeted subjects who have difficulties to answer to a designed questionnaire—children or cognitively-intact older adult [10,13,24]. In this case, an interview can be an alternative to a questionnaire.

Video analysis. To observe subjects' natural expression and body movement, 10 research analyzed video of the evaluation session recorded by external cameras or the robot. The research used an existing coding scheme to rate each video clip. Multiple researchers analyzed the recorded video and they translate it into quantified data.

Among 10 studies using a video analysis method, 6 research targeted children or the elderly. There are a number of papers on video analysis to understand various issues related to children or the elderly with dementia [29–33]. Ref. [20] found that video analysis is an appropriate way to measure users' behavioral and emotional engagement especially people with dementia.

Biometrics. There were two studies which used biometrics to evaluate social robots [34,35]. In [34], researchers collected participants' respiration, heart electrical activity, and galvanic skin response (GSR) to verify health benefit from a touch-centered interaction with a social robot. In [35], subjects' heart rate variability (HRV) from electrocardiogram (EKG) and electrodermal responses (EDR) were collected to examine the effect of robotic social therapy on children with autism. Ref. [36] found that the benefit of using biometrics is not only providing an enormous number of usability and UX issues, but also providing validation of issues, and they mentioned the potential of mixed-method approaches.

### 3.2.3. Criteria for Selecting Participants

If a robot has a specific target user, the researcher should recruit participants who meet their criteria to reflect the target market for the product. Since a social robot has potential for helping children or the elderly, 14 studies we reviewed targeted children or senior. In this case, it is hard to answer all the questions to evaluate a robot by targeted users, so some of the studies reviewed conducted an additional interview with their family, caregivers or teachers of target users. Refs. [37,38] tried to examine an effect on children with autism, so they recruited both children with autism and children in typical

development for comparison of both groups. However, most of the articles we reviewed did not have specific target users, so they recruit participants without limitation.

### 3.3. Evaluation Dimensions

To address RQ2, we will provide an overview of consumer acceptance of new technology models first. After that, we will discuss most frequently used measures from recent studies we covered. We will end with categorized measures and our own suggestions.

### 3.3.1. Consumer Acceptance of New Technology Models

After introducing a social robot as a new product category to the consumer market, one research trend was focusing on user acceptance of new technology. The traditional TAM (Technology Acceptance Model) developed by Davis suggested two major determinants of user acceptance: (1) perceived ease of use and (2) perceived usefulness [39]. TAM started with an idea that in evaluating an item or a product one of its main aims should be increasing users' job performance [40]. 3 years later, Davis published an addendum of TAM to add the third determinant of user acceptance, called perceived enjoyment [41]. Perceived enjoyment was defined as "the extent to which the activity of using the computer is perceived to be enjoyable in its own right, apart from any performance consequences that may be anticipated." [41].

Based on the revised TAM [41], a number of relevant research were published that can prove the importance of perceived enjoyment [42–44]. These research found evidence that a specific type of information system could be accepted by a user because of their perceived enjoyment. This type of system is a web-based system which users assumed it as an entertainment tool. In [45], they called this type of product hedonic product. Based on his study, a hedonic system has the value when the user experiences fun while using the system and it can prolong the time that people spend for using the system [45]. TAM [39] started with the idea that the user will need to use the information system for increasing productivity in the office environment. However, the theory of a hedonic system is more related to a product or system in the home environment, and the theory is widely accepted by many of the social robot research.

In [46], they formulated UTAUT (Unified Theory of Acceptance and Use of Technology) based upon previous user acceptance models and its core determinants did not include subjective norms such as fun or enjoyment of users. However, they revised his formula with a new determinant, hedonic motivation [47]. In this research, they assume that their innovativeness, novelty seeking and perceptions of the novelty of a target product or system will moderate the effect of hedonic motivation on behavioral intention to be affected by consumers' experience [47]. We can understand the importance of hedonic motivation from previous research on TAM and the effect of hedonic motivation in both employee aspects and consumer aspects.

A study introduced the Almere model, an acceptance model derived from UTAUT, to test variables that relate to the acceptance of social robots among elderly users [48]. The model included new determinants that are not included in UTAUT such as anxiety and attitude towards the use of technology. Experiments conducted with iCat robot verified the assumption of the researchers that social agents are both utilitarian and hedonic. After publishing the Almere model, there is much research that evaluated social robot based on this model.

### 3.3.2. Referred Evaluation Tools of Subjective Measures

Five subjective evaluation tools that were used in the articles reviewed are summarized in Table 3.

**Table 3.** Evaluation tools and main measures.

| Evaluation Tool | Main Measures |
|---|---|
| SASSI(Subjective Assessment of Speech System Interfaces) [2] | system response accuracy, likeability, cognitive demand, annoyance, habitability, speed |
| AttrakDiff/AttrakDiff 2 [3] | pragmatic quality, hedonic quality identification, hedonic quality stimulation, attraction |
| NARS(Negative Attitude toward Robots Scale) [4] | negative attitude toward situations of interaction with robots, negative attitude toward social influence of robots, negative attitude toward emotions in interaction with robots |
| Godspeedscale [5] | anthropomorphism, animacy, likeability, perceived intelligence, perceived safety |
| RoSAS(Robot Social Attributes Scale) [19] | competence, warmth, discomfort |

### 3.3.3. Most Frequently Used Measures from Recent Studies

There are 167 measures from 36 articles we reviewed after removing duplicated items. Most of the measures used are Likert scale and some are based on semantic differential techniques which included pairs of words that are opposite in meaning. Before categorizing the measured variables, we will review 3 measures which were mostly used to evaluate a social robot: (1) Enjoyment, (2) Ease of use, and (3) Trust.

Enjoyment. Refs. [9,18,23,27,37,49,50] The concept of enjoyment of use originated from the hedonic system by Van der Heijden [45] wherein it was considered as a crucial determinant for the Intention to Use. The influence of enjoyment of use on the intention to use varies depending on whether the subject is a hedonic system or a utilitarian system. According to previous research [44,47], perceived enjoyment didn't show direct influence on Intention to use, but it had an influence on ease of use and usefulness. Ref. [49] evaluated user acceptance on a huggable robot, Hugvie with three main dimensions: intention to use, enjoyment of use and ease of use. Enjoyment of use had four subordinate measures: enjoy, interested, bored, have fun. Ref. [18] found so perceived enjoyment as a factor which has an effect on the intention to use. Ref. [9] adopted Almere model and they found that enjoyment was a measure which has an influence on user acceptance. Ref. [50] evaluated several aspects of human-robot interaction, and enjoyment was a measure to evaluate the interaction between user and robots.

Ease of use. Refs. [9,17,23,28,49,51,52] Ease of use was one of the main dimensions in TAM, and it was defined as: the degree to which a person believes that using a particular system would be free of effort. TAM also suggests that the definition of "ease" is "freedom from difficulty or great effort" [39]. In UTAUT, ease of use was defined as the degree to which an innovation is expected to be free of effort [53]. Ease of use was one of the important determinants of user acceptance in UTAUT and Almere model.

Ease of use is a concept which accepted widely, and its definition is varied. Some research considered ease of use as a substitute to the concept of usability, while others defined ease of use as a sub-measure of usability. Along with ease of use, effectiveness and efficiency were also considered as sub-measures of usability. Effectiveness is a quantitative measurement that is measured using the task completion rate. Efficiency is also a quantitative measure and is defined as the task completion time. However, ease of use is a qualitative and subjective measure. Ease of use measures task-performance satisfaction [54].

Ref. [49] adopted ease of use as one of three main dimensions and ease of use had two sub-measures: easy, easily understand how to use. Ref. [17] used SUS (System usability scale) and SUS define ease of use as one of 10 sub-measures to evaluate usability. In [17], SUS score was processed to be on a scale of 0–100. Ref. [51] defined four main variables as similarity, friendship, usability, and motivation. Ease of use is was sub-measure of usability. For usability evaluation, ref. [51] used SUS and participants were asked to respond on a 5-point Likert-scale. Ref. [23] evaluated a social robot with three dimensions: attitudinal

beliefs, social normative beliefs and control beliefs. Ease of use is was a subordinate concept of attitudinal beliefs. In [52], attitudinal beliefs included both utilitarian and hedonic product aspects. It referred to UTAUT and defined ease of use as a subordinate concept of the utilitarian aspect [55]. When they evaluated its voice interaction between a robot and a user, they use ease of use as one of the metrics. Ref. [9] referred to Almere model to evaluate the human-robot interaction, and Almere model contains ease of use as a measurement of user acceptance. Ref. [28] evaluated a social robot receptionist who can speak. As [52] did, ref. [28] used ease of use to measure users' satisfaction of its interaction between a social robot receptionist and a user.

Trust. Refs. [9,16,18,23,28,51,52] In [52], they defined a concept of trust: "a belief, held by the trustor, that the trustee will act in a manner that mitigates the trustor's risk in a situation in which the trustor has put its outcomes at risk". In [18], they suggested a concept that a social robot as a viable solution to the psychometric assessment of the elderly. They referred to revised UTAUT to evaluate elderly participants' intention to accept and use this social robot. Ref. [48] defined trust as the belief in the system that the system performs the task with personal integrity and reliability. They divided the concept of trust into two items, the trust which a user has in the robot and to what extent the user intends to comply with the contents given by robots and on the nature of its interaction. In [53], trust was more important particularly in a high-risk situation and it directly affects the intention to use or acceptance of robot-decided information. Ref. [23] also pointed out that trust was an important factor to accept and use of social robots because the robot will actively take part in the decision making process. Ref. [23] used the trustworthiness scale that had 6 sub-measures of trustworthiness; honest, trustworthy, honorable, moral, ethical, genuine. Ref. [52] set two hypotheses: (1) Humor will increase trust. (2) Empathy will increase trust. To verify these, ref. [55] evaluated social robots in terms of three-way of appeal; robot appearance appeal, task appeal and content appeal. Trustworthiness was one of four measures to evaluate content appeal. It evaluated not only contents' hedonic value but pragmatic value. However, there was no significant relationship between humor-trust and also in empathy-trust in contrast with several previous research. Ref. [9] was referred to the Almere model to evaluate human-robot interaction scenario. They used trust as a measure of user acceptance. In [28], they designed exactly the same experiment with [52] to evaluate a social robot receptionist. They used trust to evaluate robot's contents. Ref. [16] considered trust as one of the important affective elements in human interaction with technology-based things. Trust was evaluated using empathy and physical touch as the measuring variables. Ref. [51] mentioned that trust is important when people accept others' recommendations. Trust is considered as an important factor in human-robot interaction because a social robot acts as an advisor in the relationship. In this study, ref. [51] reviewed previous literature to define trust and source credibility separately. Previous research which was not related to HRI, they used trust as a measure to evaluate the credibility of the contents.

### 3.3.4. Categorization of Social Robot's Evaluation Measures

Since the objective of this article was to establish evaluation dimensions for HRI research, we reviewed 167 measures from 36 articles and classified them into four main dimensions: (1) ease of use, (2) usefulness, (3) trust, and (4) emotional attractiveness. When we evaluate a social robot in the utilitarian aspect, ease of use and usefulness will be considered as important dimensions. However, when we want to evaluate a social robot in terms of hedonic aspect, emotional attractiveness is more important than ease of use or its usefulness. Finally, there is a unique dimension, trust. It is traditionally used to evaluate a media which contain and deliver contents to others and nowadays, a social robot could be one of the media that can deliver contents to users. So we found out that trust is a meaningful dimension in evaluating a social robot which needed to build a relationship with users. Those three dimensions build user's attitude toward a robot, and each of the dimensions will meet a specific goal of evaluation (Figure 2).

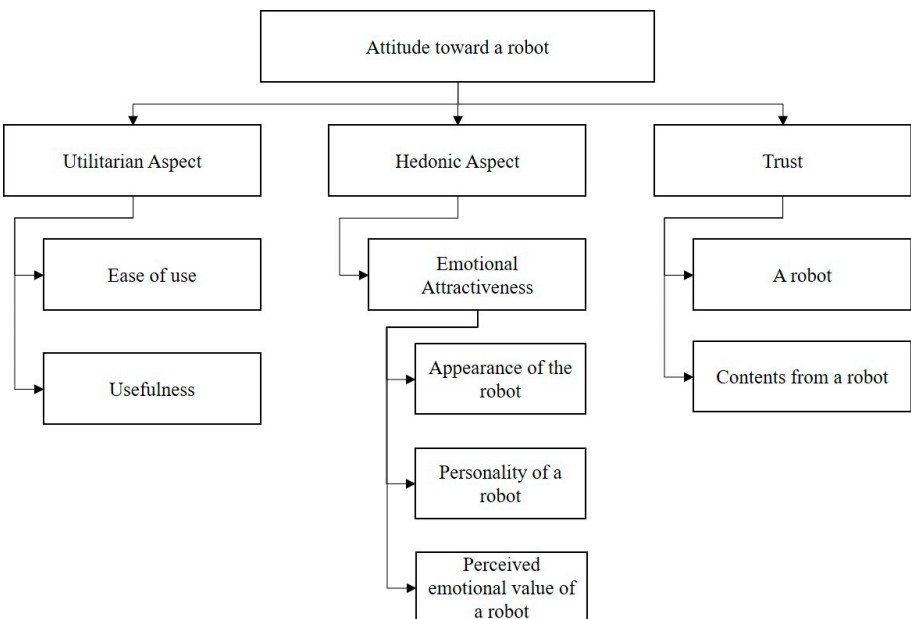

**Figure 2.** Dimensions of attitude toward a robot.

Utilitarian Aspect. There are two sub-dimensions on assessment measures in utilitarian aspect: ease of use, usefulness. Ease of use, as mentioned previous part, ease of use can be defined as "the degree to which a person believes that using a particular system would be free of effort." [39]. Ease of use was addressed in TAM, UTAUT and Almere model. Additionally, there were numerous research to define and evaluate ease of use in UX studies.

Ease of use is regarded as an important dimension in HRI aspect. When people are first exposed to a brand-new technology or product, a degree of ease of use can act as an entry barrier. People do not use a product or technology just because it is easy to use, however they will not use it consistently when they find it isn't easy to use. This element is a minimum requirement to use, and it can be a must-be quality in KANO model.

From previous UX research, they divided the task into sub-process (e.g., command, procedure, feedback and recovery) and evaluate ease of use in each process. However, a task is ambiguous when people interact with a social robot. Therefore, there are not specific constituents of ease of use in HRI studies. All of the measures of ease of use that founded from studies we reviewed are categorized in Table 4.

The second factor is usefulness. In TAM [39], perceived usefulness was defined as "the degree to which a person believes that using a particular system would enhance his or her job performance". Usefulness was traditionally a main dimension of attitude toward technology with ease of use. After defined in TAM, it is referred to Almere recently. It was a significant determinant to evaluate the functional acceptance of a utilitarian system. Social robot is a hedonic system but also a utilitarian system partly. As in the case of ease of use, a social robot should provide at least minimum acceptable usefulness to overcome a barrier of entry to the market and users. To evaluate usefulness, there are two sub-categories: (1) capability of a social robot, (2) satisfaction from a social robot's advice. However, it is difficult to categorize measures into two sub-categories based on its definition of words. In this study, we consolidate all measures of usefulness in Table 5 regardless of two sub-categories.

**Table 4.** Measures of ease-of-use.

| Measures | Count | Ratio (%) | Reference |
|---|---|---|---|
| Easy to use | 8 | 22% | [9,17,18,23,28,49,56,57] |
| Simple (Complicated) | 4 | 11% | [17,28,57,58] |
| Easy to understand how to use | 3 | 8% | [37,56,59] |
| Undemanding (Demanding, Challenging, Cumbersome) | 3 | 8% | [17,28,52] |
| In control (Out of control) | 3 | 8% | [10,28,52] |
| Need help to use | 2 | 6% | [17,58] |
| Clear to understand | 2 | 6% | [28,52] |
| Easy to learn (Hard to learn) | 1 | 3% | [17] |
| Consistent (Inconsistent) | 1 | 3% | [17] |

**Table 5.** Measures of usefulness.

| Measures | Count | Ratio (%) | Reference |
|---|---|---|---|
| Usefulness | 4 | 11% | [9,18,23,52] |
| Competent (Incompetent) | 3 | 8% | [10,51,60] |
| Responsive | 3 | 8% | [10,21,37] |
| Helpful (Unhelpful) | 2 | 6% | [28,57] |
| New (Common): Task | 2 | 6% | [28,57] |
| Knowledgeable | 2 | 6% | [10,51] |
| Informative: Contents | 2 | 6% | [28,57] |
| Related: Contents | 2 | 6% | [28,57] |
| Flexible: Contents | 2 | 6% | [28,57] |
| Functional | 1 | 3% | [17] |
| Capable | 1 | 3% | [10] |
| Expert (Inexpert) | 1 | 3% | [51] |
| Bright (Stupid) | 1 | 3% | [51] |
| Trained (Untrained) | 1 | 3% | [51] |
| Informed (Uninformed) | 1 | 3% | [51] |

Hedonic Aspect. There are three sub-dimensions on assessment measures in hedonic aspect. Overall emotional attractiveness is a main component to measure a social robot in hedonic aspect and appearance of the robot, personality of a robot, perceived emotional value of a robot are sub-dimension of emotional attractiveness.

In this study, we propose the term "emotional attractiveness" as an umbrella term to signify perceived enjoyment and users' feeling. Perceived enjoyment is referred to revised TAM [41] and it is defined as a hedonic motivation in UTAUT and Almere [46,47]. As mentioned, a social robot is a utilitarian product and a hedonic product partly, so it is defined as an important attribute of intention to use a hedonic product [45]. We categorized constituents into three concepts: (1) appearance of a robot, (2) personality of a robot, and (3) perceived emotional value of a robot (Tables 6–9).

**Table 6.** Overall attractiveness of a robot.

| Measures | Count | Ratio (%) | Reference |
|---|---|---|---|
| Attractive/Appealing/Desirable | 4 | 11% | [23,37,51,61] |
| Presentable (Unpresentable) | 2 | 6% | [28,57] |
| Inviting (Rejecting) | 2 | 6% | [28,57] |

**Table 7.** Appearance of a robot.

| Measures | Count | Ratio (%) | Reference |
|---|---|---|---|
| Anthropomorphism: Lifelike/Humanlike/Natural | 4 | 11% | [11,22,23,61] |
| Scary/Fright | 3 | 8% | [8,19,24] |
| Sad | 3 | 8% | [13,24,25] |
| Angry | 3 | 8% | [13,24,25] |
| Worried/Depressing | 2 | 6% | [34,37] |
| Lively | 2 | 6% | [37,60] |
| Organic | 1 | 3% | [19] |
| Strange | 1 | 3% | [19] |
| Dangerous | 1 | 3% | [19] |
| Upset | 1 | 3% | [38] |
| Amusing | 1 | 3% | [37] |
| Alive | 1 | 3% | [27] |
| Elegant (Rough) | 1 | 3% | [57] |
| Strong (Weak) | 1 | 3% | [57] |
| Tense | 1 | 3% | [61] |

**Table 8.** Personality of a robot.

| Measures | Count | Ratio (%) | Reference |
|---|---|---|---|
| Actively engaged | 7 | 19% | [20,28,57,58,60–62] |
| Nice/Kind/Good (Awful/Unkind/Bad) | 6 | 17% | [11,19,28,57,61,63] |
| Confident (Insecure) | 4 | 11% | [10,17,28,57] |
| At ease/Relaxed/Calm | 4 | 11% | [20,28,34,57] |
| Sociable (Unsociable) | 3 | 8% | [18,23,61] |
| Aggressive/Offensive | 3 | 8% | [13,19,37] |
| Interactive | 3 | 8% | [13,19,51], |
| Companionship/As a co-worker (Bossy) | 3 | 8% | [13,23,51] |
| Perceived emotional stability/Insensitive (Sensitive) | 2 | 6% | [22,61] |
| Independent (Dependent) | 2 | 6% | [10,16] |
| Exciting (Lame) | 2 | 6% | [57,61] |
| Sympathetic (Unsympathetic) | 2 | 6% | [51,60] |
| Receptive | 1 | 3% | [28] |
| Conscious (Unconscious) | 1 | 3% | [11] |
| Perceived pet likeness | 1 | 3% | [22] |
| Extrovert (Introvert) | 1 | 3% | [57] |
| Rational (Emotional) | 1 | 3% | [57] |
| Familiarity | 1 | 3% | [27] |
| Sincere | 1 | 3% | [51] |
| Shy | 1 | 3% | [13] |

**Table 9.** Perceived emotional value of a robot.

| Measures | Count | Ratio (%) | Reference |
|---|---|---|---|
| Pleasant (Unpleasant) | 9 | 25% | [11,19,20,25–28,57,61] |
| Friendly/Could be a friend (Unfriendly) | 8 | 22% | [11,13,22,28,37,51,57,61] |
| Anxiety towards a robot | 4 | 11% | [9,18,20,23] |
| Happy | 4 | 11% | [13,19,24,25] |
| Satisfied (Frustrated) | 4 | 11% | [28,34,57,61] |
| Close/Connected (Distant) | 3 | 8% | [16,21,61] |
| Empathetic (Not empathetic) | 2 | 6% | [16,57] |
| Friendly communicative | 1 | 3% | [60] |
| Compassionate | 1 | 3% | [19] |
| Stimulating | 1 | 3% | [37] |
| Surprise | 1 | 3% | [24] |
| Can spend a good time with | 1 | 3% | [61] |
| Entertaining | 1 | 3% | [61] |
| Pleasant (Unpleasant) | 9 | 25% | [11,19,20,25–28,57,61] |
| Friendly/Could be a friend (Unfriendly) | 8 | 22% | [11,13,22,28,37,51,57,61] |
| Compassionate | 1 | 3% | [19] |
| Stimulating | 1 | 3% | [37] |
| Surprise | 1 | 3% | [24] |
| Can spend a good time with | 1 | 3% | [61] |
| Entertaining | 1 | 3% | [61] |

Trust. Trust is defined as "the belief that the system performs with personal integrity and reliability" in Almere model [48]. The studies related to relational agent indicate that trustworthy is the main dimension to evaluate the agent [55,64]. After this research published, Almere model also used trust as one of its constructs [48]. Based on trust in a social robot as an agent, the user can accept a piece of advice from a robot [51]. Consequently, there can be constant interaction with a robot and build a solid foundation for a relationship between a user and a robot.

In [52], there are two individuals that can define trust—a trustor and a trustee. The trustor is an individual who is exposed to risk. The trustee can represent the other one who can give trust to the trustor. In the relationship between a user and a social robot, a user who will need help can be a trustor and then a social robot will play an opposite role as a trustee. Based on these relationship dynamics, we divided measures of trust into two groups: (1) trustworthiness of a robot, (2) acceptance toward an advice from a robot. However, we only collected a few measures of trust from previous researches and those are in Table 10.

**Table 10.** Measures of trust.

| Measures | Count | Ratio (%) | Reference |
|---|---|---|---|
| Trustworthy/Credible/Complianc | 11 | 31% | [9,10,16,18,19,21,23,28,51,57,62] |
| Intelligent (Unintelligent) | 3 | 8% | [23,51,61] |
| Virtuous (Sinful) | 1 | 3% | [51] |
| Unselfish (Selfish) | 1 | 3% | [51] |

## 4. Discussion

Since the objective of this study was to propose pragmatic suggestion of measures to answer the research requirements, we investigated the evaluation methodology applied among various research and derived three core-constructs of attitude towards a social robot. Based on this result, we suggest evaluation methods as follows.

### 4.1. Proper Combination of Stimuli and Evaluation Techniques

When a researcher use questionnaire to evaluate, they can give any type of four stimuli for the user. Text, image, video and live interaction can all convey information to complete a questionnaire. We suggest the same type of stimuli for the interview. However, when a study will analyze biometric or video, live interaction would be best stimuli for evaluation. Passive interaction such as reading text or watching image and video is not sufficient to observe changes of biometrics or user's behavior.

### 4.2. Suggestion of Stimuli and Evaluation Technique by Assessment Conditions

First, this study suggests appropriate evaluation stimuli according to the dimension of social robot assessment. Utilitarian robot and hedonic robot needed a different approach. When evaluating a utilitarian robot, participants should focus on its performance and function. Text, video and live interaction can show its function but the static image is insufficient to deliver the whole process of conducting the task. However, when the user evaluate hedonic robot, they will focus on its emotional value. A static image can describe robots appearance and facial expression better than text. Regarding concept of trust, it is difficult to illustrate the expected intellectual capacity of a robot in text or image. Therefore, video or live interaction would be recommended to assess trust between robot and user.

Next, proper evaluation techniques according to the dimension of social robot assessment are proffered. Questionnaire and interview can cover all three dimensions of attitude towards a robot. In contrast, measuring biometric is only recommended for assessment in hedonic aspect. Biometric can be translated emotional arousal and valence into numbers, and it is applicable to evaluate a social robot in hedonic aspect. Last, video analysis is suitable for evaluation in utilitarian or hedonic aspects. Trust can be observed in a long-term relationship between robot and users, and it is insufficient to measure trust by biometric or video analysis. Measuring trust is needed deep dive into users and in-depth interview is the most appropriate way to evaluate trust in robot.

To provide proper UX evaluation methodology and measures, ref. [65] was a good reference for this study. Ref. [65] propose new point of view to evaluate HRI, users' possibilities to recognize a robot's action and intention.

They suggested UX evaluation process for HRI from preparation to identifying UX problem thoroughly. The overall procedure is described in detail for everyone and pointed out a meaningful issue that importance of defining context to set product goal and select appropriate UX evaluation type.

They mentioned that the developmental stage of the product is one of the criteria to select UX evaluation type. It is important to find out the ecological problem of a product early in the development cycle, and even 'quick and dirty' evaluation is needed at that time. In this paper, we also indicate the importance of problem finding in the early stage of the development cycle and suggest an applicable type of evaluation stimuli and evaluation methodology for that.

Additionally, they stressed the advantage of field study as an evaluation method. When evaluating a robot in a lab setting, sometimes we can observe that users are self-conscious and do unnatural behavior. So they said gathering more accurate data from multiple evaluation methodology will be complementary. We also agree with this point, and we put together the evaluation context of reference paper in table [11] and hope it could be helpful to select an evaluation method for practitioners.

However, ANEMONE focused on the procedure of evaluation, not the evaluation measure level. They provided some examples of objective and subjective UX measure,

but presented examples are not sufficient to design whole UX evaluation measure and its structure for practitioners. In this study, we try to provide as much measure as we can for practitioners to design UX evaluation easily. Also three categories we suggested as utilitarian aspect, hedonic aspect and trust could be a valuable foundation to create the base structure for the assessment of attitude toward a robot.

Ref. [54] was also a good starting point of this study to understand the latest issues of interaction between human and social robots. Ref. [54] analyzed four trends and three challenges related to UX of HRI. These issues would be a useful basis for finding out the proper topic of future research. However, they didn't offer a specific guideline of evaluation methodology or measures.

One of meaningful argument we found from their study was the necessity of robot developers' knowledge of UX evaluation. To develop a social robot successfully, developers should understand which elements can affect the UX and can make positive feelings for intended users. We totally agree with this point of view and the aim of this paper is providing a practical guideline to UX evaluation in HRI for everyone. Those evaluation measures we collected can make people think about the various attitude toward a robot that the intended user can have. Especially developers can make the relation between those attitude keywords to product elements such as duration of specific motion, the pitch of robot's sound. We hope that it will provide the foundation for commonly agreed structure and measures to develop and evaluate UX of HRI.

## 5. Conclusions

It is against this backdrop of rapid technological advancements as well as rising interests on social robots that we have launched a research effort to collect and reorganize evaluation methodology and measures. Practitioners make effort to find the important value of social robots for the consumers to expand social robots market and academics keep studying methodology to evaluate social robots and evaluation indices for improvement and refinement existing social robots. This paper is expected to give benefits as below:

1.  For planning and designing new social robots

    -   Chance to consider significant and affecting factors for designing social robots from the planning phase of development

2.  For evaluation of new social robot

    -   Helps to design survey questionnaires easily from the whole set of evaluation measures previously used

However, there is still a limitation to this research. There was no chance to validate this structure we've suggested. It is concern point that whole set of evaluation measures we've collected from each research can be not significant. These measures came from various research and could be only working as a subset and meaningful for each research paper. However, they might not work as it was because of changes in structures of evaluation measures.

Although we believe that offering the whole set of evaluation measures is meaningful for readers to help to select an appropriate subset from measures pool we've collected at this point. To make this research more valuable, further research to categorize social robots, design appropriate evaluation measures structure for each category and validate each structure is needed.

With an astonishing growth of interest on social robots, it is expected that this research can be a good starting point for many academics and practitioners who studying social robots and can also contribute to market expansion.

**Author Contributions:** Conceptualization, M.J.; methodology, M.J.; formal analysis, M.J.; investigation, M.J. and M.J.S.L.; resources, M.J., M.J.S.L. and M.H.Y.; data curation, M.J.; writing—original draft preparation, M.J. and M.J.S.L.; writing—review and editing, M.J.S.L.; visualization, M.J. and M.J.S.L.; supervision, M.J.S.L. and M.H.Y.; All authors have read and agreed to the published version of the manuscript.

**Funding:** This research received no external funding.

**Institutional Review Board Statement:** Not applicable.

**Informed Consent Statement:** Not applicable.

**Conflicts of Interest:** The authors declare no conflict of interest.

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
