# Peer review of "Evaluation of Methodologies and Measures on the Usability of Social Robots: A Systematic Review"

_applsci, doi:10.3390/app11041388_

Round 1

Reviewer 1 Report

The paper is clearly organized, self contained and presented. However, Its main weakness is that its novelty is low. Therefore, while technically correct, I do not consider that this work has sufficient novelty to recommend its publication. I encourage the author to explain the advantages and gains in his/her own work over other’s works more.

Author Response

Comment 1-1

The paper is clearly organized, self contained and presented. However, Its main weakness is that its novelty is low. Therefore, while technically correct, I do not consider that this work has sufficient novelty to recommend its publication.

Response 1-1

à Comment accepted and added additional explanation and content in the introduction section.

“ Robot industry has been steadily growing since after the first industrial robots developed in 1960s. At that time, the most appealing role of robots is replacing human beings to do unreachable by or unsafe job [57].   For those kinds of robots, there are clear evaluation indices like task completion rate and time. However, we can’t apply those indices for evaluation of social robots. When it comes to social robot, their task is ambiguous and everyone has different expectations for the role of that. We still can apply traditional evaluation indices in HCI such as ease of use, usefulness. However, usability is basic concept and further research is needed on what makes social robots attractive. In this paper, we will suggest main pillars affecting attitude towards social robots, and readers can easily understand what’s important in designing a social robot to make it more attractive.

One more thing that is difference from the traditional HCI research is evaluation methodology. In the traditional HCI research area, most of elements that we want to evaluate are visual components and there are various prototyping tools to make visual stimuli easily for quick and dirty research. However, it is still difficult to develop working prototype which can be executed like final commercial product and practitioners constantly needed ideas how to evaluate social robots in early stage of developing process.   In order to  help them , our analysis is guided by these following research questions.”

Additional Explanation 1-1

--> This paper proposed a new structure of evaluation measures for social robots collected in previous researches. When it comes to evaluation measures, we can’t say these are unique and novel. However, we believe this research can contribute to the social robot research area. This paper introduces well organized structure of evaluation measures and this structure can guarantee its generality unlike fragmented evaluation indices that are used to evaluate specific target robots only for each paper with purpose. Moreover, this research can be a practical basis for further study. Social robots can be categorized based on its characteristics, and appropriate evaluation measures for each category can be derived from this collected measures.

Comment 1-2

I encourage the author to explain the advantages and gains in his/her own work over other’s works more.

Response 1-2

à Comment accepted and additional explanation is added in the discussion secion.

“This paper can give practical help
1. For planning and designing new social robot
- Can consider significant factors influencing on attitude towards social robots at the beginning of product development process
2. For evaluation of new social robot
- Can design survey index easily from the whole set of index which are previously used and thoroughly validated”

Additional Explanation 1-2

à It is easy to find papers that evaluate specific social robot with designed evaluation index with specific purpose. However there aren’t enough research that contain overall evaluation measures collected from previous research and reorganize these measures thoroughly. When we look into each paper that evaluates a social robot, usually they use evaluation index which are appropriate for the characteristics of evaluation target robot. So if researchers analyze a few papers as references to find out evaluation index for their research, we can’t guarantee that they found complete set of evaluation measures that can cover most of social robot types. That was the starting point of our research and we believe this paper can contribute significantly due to followings.

Additional Notes:

            This paper is subjected to professional English editing service upon acceptance.

Reviewer 2 Report

The paper considers interesting topic as it is social robots with their main goal to provide meaningful social interaction with human.  It is a very up-to-date topic since robotic technology is growing rapidly and social robot are expected to fill a lack of human resources. Social robot as a new type of personal service robots are meant to support people with special needs and aged population (eldery people).

The main objective of the study was to analyze methods of evaluating the usability of social robots deployed in academia based on traditional usability testing methodologies and make it more accessible and systematic to practitioners in the relevant industry, to help them to find out the most appropriate methodology to evaluate a social robot with specific requirements.

To achieve the main aim of the study the analysis was guided by two following research questions:

  • What kind of usability evaluation methodologies are deployed in social robot studies?
  • What are the main evaluation dimensions in social robot studies?

Did the aim of the research arise from conducted literature analysis or it refers to pracitical needs? Please explain it more detailed. The introduction chapter should also discuss the need of developing some practical strategies for selecting appropriate methods and measures. There is no indication why such research should be undertaken

The authors investigated the evaluation methodology applied among various research and derived three core-constructs of attitude towards a social robot. Based on this result, they suggested evaluation methods and practical strategies for selecting appropriate methods and measures that meet specific requirements of research.

Future research challenges and areas were indicated.

Are there any research limitations? Please formulate them.

The conclusions were formulated quite superficially and laconically, so they need to be supplemented and extended.

The paper was carefully edited. The English language and style are fine.

I find the solution proposed by the authors positively: interesting and useful results were accomplished, proper methodology and tools allowing for achieving goals were involved, adequate conclusions were formulated.

I recommend this paper to be published after minor corrections.

Author Response

Comment 2-1

The paper considers interesting topic as it is social robots with their main goal to provide meaningful social interaction with human.  It is a very up-to-date topic since robotic technology is growing rapidly and social robot are expected to fill a lack of human resources. Social robot as a new type of personal service robots are meant to support people with special needs and aged population (eldery people).

The main objective of the study was to analyze methods of evaluating the usability of social robots deployed in academia based on traditional usability testing methodologies and make it more accessible and systematic to practitioners in the relevant industry, to help them to find out the most appropriate methodology to evaluate a social robot with specific requirements.

To achieve the main aim of the study the analysis was guided by two following research questions:

What kind of usability evaluation methodologies are deployed in social robot studies?

What are the main evaluation dimensions in social robot studies?

Did the aim of the research arise from conducted literature analysis or it refers to pracitical needs? Please explain it more detailed.

Response 2-1:

à comments accepted and additional explanation added in the introduction section.

“ Robot industry has been steadily growing since after the first industrial robots developed in 1960s. At that time, the most appealing role of robots is replacing human beings to do unreachable by or unsafe job.  For those kinds of robots, there are clear evaluation indices like task completion rate and time. However, we can’t apply those indices for evaluation of social robots. When it comes to social robot, their task is ambiguous and everyone has different expectations for the role of that. We still can apply traditional evaluation indices in HCI such as ease of use, usefulness. However, usability is basic concept and further research is needed on what makes social robots attractive. In this paper, we will suggest main pillars affecting attitude towards social robots, and readers can easily understand what’s important in designing a social robot to make it more attractive.

One more thing that is difference from the traditional HCI research is evaluation methodology. In the traditional HCI research area, most of elements that we want to evaluate are visual components and there are various prototyping tools to make visual stimuli easily for quick and dirty research. However, it is still difficult to develop working prototype which can be executed like final commercial product and practitioners constantly needed ideas how to evaluate social robots in early stage of developing process.   In order to  help them , our analysis is guided by these following research questions.”

Additional Explanation 2-1

à Working as a designer, I was frequently asked those questions from my colleagues who are designing a personalized companion robot such as Ballie. However, as we mentioned previously, there are hardly any previous research that can answer those questions. There were only fragmented researches that evaluate specific social robots. So we can tell that these research questions have arisen from practical needs first, and literature analysis supported it additionally.

Comment 2-2

The introduction chapter should also discuss the need of developing some practical strategies for selecting appropriate methods and measures. There is no indication why such research should be undertaken

à To answer this question, we’ve added following paragraphs in the introduction.

“Robot industry has been steadily growing since after the first industrial robots developed in 1960s. At that time, the most appealing role of robots is replacing human beings to do unreachable by or unsafe job. For those kinds of robots, there are clear evaluation indices like task completion rate and time. However, we can’t apply those indices for evaluation of social robots. When it comes to social robot, their task is ambiguous and everyone has different expectations for the role of that. We still can apply traditional evaluation indices in HCI such as ease of use, usefulness. However, usability is basic concept and further research is needed on what makes social robots attractive. In this paper, we will suggest main pillars affecting attitude towards social robots, and readers can easily understand what’s important in designing a social robot to make it more attractive.

One more thing that is difference from the traditional HCI research is evaluation methodology. In the traditional HCI research area, most of elements that we want to evaluate are visual components and there are various prototyping tools to make visual stimuli easily for quick and dirty research. However, it is still difficult to develop working prototype which can be executed like final commercial product and practitioners constantly needed ideas how to evaluate social robots in early stage of developing process.”

Comment 2-3

The authors investigated the evaluation methodology applied among various research and derived three core-constructs of attitude towards a social robot. Based on this result, they suggested evaluation methods and practical strategies for selecting appropriate methods and measures that meet specific requirements of research.

Future research challenges and areas were indicated.

Are there any research limitations? Please formulate them.

à Comment accepted and revised the dicussion section.

“There was no chance to validate this structure we’ve suggest. It is concern point that whole set of evaluation measures we’ve collected from each research can be not significant. These measures came from various research, and could be only working as a subset and meaningful for the each research paper. However they might not work as it was because of changes in structures of evaluation measures.

Although we believe that offering the whole set of evaluation measures is meaningful for readers to help selecting appropriate subset from indices pool we’ve collected at this point. To make this research more valuable, further research to categorize social robots, design appropriate evaluation measures structure for each category and validate each structure is needed.”

Additional explanation 2-3

à In this paper, we gathered various evaluation measures from previous researches, and reorganize these systemically to suggest new structure that can explain factors affecting attitude towards social robots. However, there is still limitation in this research. We’ve added research limitations based on your comment in the conclusion. Please refer to following.

Comment 2-4

The conclusions were formulated quite superficially and laconically, so they need to be supplemented and extended.

à Comment accepted and the conclusion is revised as the following.

“It is against this backdrop of rapid technological advancements as well as rising interests on social robot that we have launched a research effort to collect and reorganize evaluation methodology and measures. Practitioners make effect to find the important value of social robots for the consumers to expand social robots market and academics keep studying methodology to evaluate social robots and evaluation indices for improvement and refinement existing social robots. This paper is expected to give benefits as below:

  1. For planning and designing new social robot

- Chance to consider significant and affecting factors for designing social robots from the planning phase of development

  1. For evaluation of new social robot

- Helps to design survey questionnaires easily from the whole set of evaluation measures previously used

However, there is still limitation in this research. There was no chance to validate this structure we’ve suggest. It is concern point that whole set of evaluation measures we’ve collected from each research can be not significant. These measures came from various research, and could be only working as a subset and meaningful for the each research paper. However they might not work as it was because of changes in structures of evaluation measures.

Although we believe that offering the whole set of evaluation measures is meaningful for readers to help selecting appropriate subset from measures pool we’ve collected at this point. To make this research more valuable, further research to categorize social robots, design appropriate evaluation measures structure for each category and validate each structure is needed

With an astonishing growth of interest on social robots, it is expected that this research can be a good starting point for many academics and practitioners who studying social robots and can also contribute market expansion”

Comment 2-5

The paper was carefully edited. The English language and style are fine.

I find the solution proposed by the authors positively: interesting and useful results were accomplished, proper methodology and tools allowing for achieving goals were involved, adequate conclusions were formulated.

I recommend this paper to be published after minor corrections.

Response to comment 2-5

à Comment accepted and appreciated.

Round 2

Reviewer 1 Report

This revised version of the manuscript now provides enough explanation of the novel aspects in HRI evaluation studies, which were absent in the former version. According to changes introduced, the paper contributes to the field of HRI evaluation studies, and provides enough support to validate the proposed data extraction and synthesis. In its current form, the paper has a fair interest to the research community and is clearly organized and presented.

Round 3

Reviewer 1 Report

This revised version of the manuscript now provides enough explanation of the novel aspects in HRI evaluation studies, which were absent in the former version. According to changes introduced, the paper contributes to the field of HRI evaluation studies, and provides enough support to validate the proposed data extraction and synthesis. In its current form, the paper has a fair interest to the research community and is clearly organized and presented.